# Minimum Quantity Lubrication Jet Noise: Passive Control

**DOI:** 10.3390/mi14101814

**Published:** 2023-09-22

**Authors:** Xiaodong Hu, Junhao Yu, Yuanlong Li, Yu Xia, Xuefeng Xu, Ruochong Zhang

**Affiliations:** 1College of Mechanical Engineering, Zhejiang University of Technology, Hangzhou 310023, China; hooxoodoo@zjut.edu.cn (X.H.); 2112102119@zjut.edu.cn (J.Y.); 17863658800@163.com (Y.L.); 1112102004@zjut.edu.cn (Y.X.); xuxuefeng@zjut.edu.cn (X.X.); 2Key Laboratory of Special Purpose Equipment and Advanced Processing Technology, Ministry of Education and Zhejiang Province, Zhejiang University of Technology, Hangzhou 310023, China

**Keywords:** jet noise, passive control, micro-groove, minimum quantity lubrication

## Abstract

Jet noise is a common problem in minimum quantity lubrication (MQL) technology. This should be given great attention because of its serious impacts on the physical and mental health of the operators. In this study, a micro-grooved nozzle is proposed based on the noise reduction concept of biological micro-grooves. The flow field and acoustic characteristics of an original nozzle and a micro-grooved nozzle were investigated numerically to help better understand the noise reduction mechanism. The reasons for noise generation and the effects of the length (*L*), width (*W*) and depth (*δ*) of the micro-grooves on noise reduction were analyzed. It was found that jet noise is generated by the large-scale vortex ring structure and the pressure fluctuations caused by its motion. The overall sound pressure level (OASPL) decreased with the increases in *W* and *δ*, and increased with the increase in *L*. Among of them, *δ* has the greatest effect on noise reduction. The maximum noise reduction achieved was 6.66 dB, as verified by the OASPL test. Finally, the noise reduction mechanism was discussed in terms of the flow field, vorticity and the frequency characteristics. Micro-grooves can enhance the mixing of airflow inside the nozzle and accelerate the process of large-scale vortices breaking into smaller-scale vortices. It also reduces the sound pressure level (SPL) of middle frequencies, as well as the SPL of high frequencies on specific angles.

## 1. Introduction

Jet noise is a common problem in industrial production. Its research was initiated by James Wright Hill’s theory of acoustic analogy, but there is still no complete theory to explain it [1]. Minimum quantity lubrication (MQL), a typical green cooling lubrication method, has the characteristics of being eco-friendly, reducing the cutting fluid amount and cutting forces, prolonging tool life and improving workpiece surface quality [2,3,4,5]. However, this method brings about a new annoying problem of aeroacoustic noise generated from its spray. The air jet mixes with the ambient atmosphere intensely to create strong jet noise. This kind of jet noise is commonly higher than 95 dB. Operators working for long periods in such an environment will not only damage their auditory systems, but their mood, sleep and work efficiency will be affected. More seriously, jet noise can induce hypertension and coronary heart disease in operators, and even have adverse effects on their nervous, digestive and reproductive systems [6,7,8]. Therefore, it is important to study the generation and noise reduction mechanisms of jet noise and develop noise reduction nozzles to improve MQL technology.

Many noise reduction techniques have been investigated to reduce jet noise. It is an effective noise reduction method to reduce the intensity of vortices by designing microstructures on the surface of the channel. The micro-grooved surface, inspired by shark skin, has been widely acknowledged as one of the most promising passive drag reduction techniques due to its simple structure, effective drag reduction and easy installation [9,10]. Micro-grooved surfaces have great potential to reduce jet noise [11]. Inspired by the micro-grooved structure of shark skin, Wang and Nakata et al. [12] introduced the grooves into the design of a mixed flow fan. They found that the groove structure can modify the blade surface flow and create vortices at the leading edge of the groove, which reduces the turbulent kinetic energy and suppresses the noise. Dang et al. [13] provided a new idea of a flow-induced noise reduction design with spanwise micro-grooved surfaces. The cause of noise reduction is due to the secondary vortex generated in the microgrooves, which hinders the process of turbulence, consumes the energy of the flow, and weakens the intensity of turbulent burst. Dai et al. [14] extracted shark skin features and established a centrifugal pump with V-grooved surface blades. The blade groove surface can change the vortex structure in the impeller flow channel, breaking up the large vortices into smaller vortices, reducing the degree of turbulence inside the centrifugal pump impeller, and reduce the acoustic power in the flow channel. Vishnu et al. [15] investigated the aerodynamic characteristics of grooved nozzles and found that grooved nozzles exhibit better mixing characteristics, shorter core lengths and faster jet decay. Xavier and Iyappan et al. [16] investigated the effect of mixing flow exiting a supersonic jet by introducing semi-circular shaped internal grooves and square shaped internal grooves on the top and bottom sides of the divergent section of the nozzle. Square grooves have better mixing and faster jet core decay than semi-circular grooves. The reason is that the sharp corners produce streamwise counter-rotating vortices which help in mass entrainment.

With the inspiration of micro-groove noise reduction concepts, in this research, a micro-groove-based noise reduction method for an MQL nozzle was developed. The purpose is to reduce the jet noise of an MQL spray without changing the original function of the nozzle. The flow field and acoustical characteristics of the original nozzle and micro-grooved nozzle were investigated using the hybrid RNAS/LES method and the Ffowcs Williams and Hawkings equation. The causes of jet noise from the nozzle and the effect of groove parameters on the noise reduction were analyzed. The noise reduction mechanism of the micro-grooves was discussed in terms of the flow field, the vortex field and the frequency spectra.

## 2. Materials and Methods

### 2.1. Original Nozzle

As shown in Figure 1, the original nozzle consists of two parts: the nozzle head and the nozzle tube. The nozzle tube transports MQL oil, and its periphery annular channel was designed for compressed air. Compressed air atomizes the lubricating fluid into micron-sized droplets, and sprays them into the cutting region to provide cooling and lubrication. It was experimentally confirmed that the droplets themselves have little effect on the noise generation during the atomization process. Therefore, the model was simplified when simulating the flow field and acoustical characteristics, and only the noise generated by the airflow was considered. The parameters of the original nozzle model are shown in Table 1.

### 2.2. Micro-Grooved Nozzle Design

Figure 2 provides an illustration of the micro-grooved nozzle structure in detail. Since micro-grooves facilitate noise control, four grooves were designed at the end of the nozzle tube with a uniform distribution. In order to investigate the effects of individual groove parameters on jet noise, a single-factor experimental study was carried out. Guo [17] investigated the effects of groove parameters on gas flow. His research discovered that the longer the groove, the less helpful it is to vortex breakdown, but increasing the width and depth of the groove contributed to vortex breakup. The initial values of the groove parameters (*L*, *W* and *δ*) in this study were selected based on the results of Guo’s study (i.e., 25% of the nozzle tube length, 15% of the nozzle tube diameter and 40% of the nozzle tube wall thickness). Considering the machining factors, *L* was selected in three values, 1 mm, 2 mm and 3 mm. If *W* and *δ* were too large, the connection between the two grooves and the wall thickness of the tube core become thin, which will affect the lifetime of the tube core. *W* was selected in three values, 0.5 mm, 0.7 mm and 1 mm. *δ* was selected in three values, 0.2 mm, 0.35 mm and 0.5 mm. These values were combined to obtain seven nozzles, and the parameters are summarized in Table 2. The actual nozzles are shown in Figure 3.

### 2.3. Experimental Setup

The sound level meter (AR844, SMART SENSOR Holding Co., Ltd., Shanghai, China) was placed on a circle of radius *R* (500 mm), which is centered on the nozzle exit. The sound level meter and the nozzle were placed 500 mm off the ground to minimize the effect of reflected sound waves. The experimental setup is shown in Figure 4. Multiple monitoring angles (*θ*) and directions were selected for the OASPL experiments due to the directivities of the aerodynamic field [18]. *θ* is the angle between the horizontal line of the nozzle exit and the line of the nozzle and sound level meter. The *θ* values selected were 15°, 30°, 45°, 60°, 75° and 90° to obtain six OASPL monitoring points [19]. *R* and *θ* were determined with a laser distance mater (Vchon Electronics Co., Ltd., Bengbu, China). During the experiment, the inlet air pressure was set to 0.3 MPa, and the OASPL was tested at six monitoring points. Each test was repeated eight times, and the average value was obtained as the final result after removing the maximum and minimum values.

### 2.4. Numerical Simulation

The spray flow field and aerodynamic acoustic field were simulated using a hybrid RNAS/LES and FW—H acoustic analogy method. The calculation region is shown in Figure 5. The calculation region outside the nozzle was 60 D in length, with widths of 10 D upstream and 20 D downstream. The sound integration surface was 10–30 D in width and 50 D in length. The flow field of the jet was first calculated numerically, and then the flow field data on the acoustic integration surface were extracted to calculate the aeroacoustic field via the FW–H equation. The inlet of the nozzle was specified as a pressure inlet with total pressure P_total_ = 0.3 MPa, while the outlet was specified as a pressure outlet boundary with relative static pressure P_0_ = 0 MPa. The nozzle wall was specified as a no-slip, adiabatic solid wall surface.

#### 2.4.1. Calculation Methods and Governing Equations

The jet flow field was first calculated in steady state to obtain the initial steady-state flow field. The steady state turbulence calculations were carried out in the realizable k–ε turbulence model [20]. Then, the transient calculations were carried out in LES turbulence model and the jet noise field was calculated by the Ffowcs Williams and Hawkings equation. The flow of the jet is compressible flow. Therefore, the Favre-filtered Navier–Stokes equation was chosen for the governing equation [21,22,23]:(1)$$\frac{\partial \bar{\rho }}{\partial t}+\frac{\partial \bar{\rho }{\tilde{u}}_{i}}{\partial x_{i}}=0$$
(2)$$\frac{\partial \bar{\rho }{\tilde{u}}_{i}}{\partial t}+\frac{\partial \bar{\rho }{\tilde{u}}_{i}{\tilde{u}}_{j}}{\partial x_{j}}+\frac{\partial \bar{p}}{\partial x_{i}}-\frac{\partial {\tilde{\sigma }}_{ij}}{\partial x_{j}}=-\frac{\partial \tau_{ij}}{\partial x_{j}}+\frac{\partial }{\partial x_{j}}\left(\bar{\sigma_{ij}}-\tilde{\sigma_{ij}}\right)$$
(3)$$\frac{\partial \hat{e}}{\partial t}+\frac{\partial \left(\hat{e}{\tilde{u}}_{i}+\bar{p}{\tilde{u}}_{i}\right)}{\partial x_{i}}-\frac{\partial {\tilde{\sigma }}_{i\;j}{\tilde{u}}_{i}}{\partial x_{j}}+\frac{\partial {\tilde{q}}_{i}}{\partial x_{i}}=RHS_{E}$$
where “$\bar{\rho }$” is a spatial-filtered variable, “$\tilde{u}$” is a mass-weighted change of variable (Favre). ${\tilde{u}}_{i}$, *ρ* and *p* are the velocity component, the fluid density and the pressure, respectively. The tensor $\bar{\sigma_{ij}}$ is the filtered stress tensor, and the computable stress tensor $\tilde{\sigma_{ij}}$ is defined as follows:(4)$$\tilde{\sigma_{ij}}=\mu \left(\tilde{T}\right)\left(\frac{\partial {\tilde{u}}_{i}}{\partial x_{j}}+\frac{\partial {\tilde{u}}_{j}}{\partial x_{i}}-\frac{2}{3}\frac{\partial {\tilde{u}}_{k}}{\partial x_{k}}\delta_{ij}\right)$$

The sub-grid scale tensor $\tau_{ij}$ is defined as follows:(5)$$\tau_{ij}=\bar{\rho }\left(\tilde{u_{i}u_{j}}-{\tilde{u}}_{i}{\tilde{u}}_{j}\right)$$

The computable heat flux vector ${\tilde{q}}_{i}$ is given by the following:(6)$${\tilde{q}}_{i}=-\frac{\mu \left(\tilde{T}\right)C_{P}}{\mathrm{Pr}}\frac{\partial \tilde{T}}{\partial x_{i}}$$
where Pr is the Prandtl number and $C_{P}$ is the specific heat. The filtered energy equation $\hat{e}$ is defined as follows:(7)$$\hat{e}=\frac{\bar{p}}{\gamma -1}+\frac{1}{2}\bar{\rho }{\tilde{u}}_{i}{\tilde{u}}_{i}$$

${\mathrm{RHS}}_{E}$ consists of 7 terms:(8)$${\mathrm{RHS}}_{E}=-B_{1}-B_{2}-B_{3}+B_{4}+B_{5}+B_{6}-B_{7}$$

The sub-grid terms $B_{1}$ to $B_{7}$ are introduced by Larchevêque [24]. To close the set of equations, the following filtered equation of state is used:(9)$$\tilde{p}=\bar{\rho }R\tilde{T}$$
where $R=287.1\;J\cdot {\mathrm{Kg}}^{-1}\cdot K^{-1}$.

#### 2.4.2. Numerical Mesh

The structured mesh was used in the calculation region, as shown in Figure 6. In order to keep the accuracy of the calculation results, the mesh was refined at the nozzle and nozzle exit, and a boundary layer was set at the contact area between the flow channel and the inner wall of the nozzle. The center region was meshed with an O-block mesh to increase the mesh quality and reduce calculation errors.

Prior to simulating the current nozzle, it is important to reduce the effect of the grid number on the simulation results. The number of boundary layers was an important factor affecting the number of meshes. To investigate the effect of grid number on the simulation results, the boundary layer numbers of 35, 40, 60 and 80 were used. The grid numbers for the four-boundary layer were 2.27 million, 2.89 million, 3.67 million, and 4.59 million. The OASPL results of the simulation and experiments were compared, as shown in Table 3. It can be observed that the simulation results generally agreed well with the experimental data when the number of grids was 2.89 million. In order to ensure the comparability of the calculation results, the mesh distribution of the micro-grooved nozzle was kept consistent with the original nozzle, except for the groove position.

## 3. Results and Discussion

### 3.1. OASPL Results

The OASPL test results of micro-grooved nozzles 1–7 and the original nozzle are shown in Figure 7. As shown in the figure, the micro-grooves have a positive effect on jet noise reduction, and the low angles (15°, 30°, 45°) are better than the high angles (60°, 75°, 90°). Compared with the original nozzle, the best noise reduction is obtained at 30°, with a reduction of 3.2% to 6.8%. In addition, micro-grooved nozzle 7 has the best noise reduction effect. An error analysis was performed on the simulated results of the OASPL, as shown in Table 4. The average relative error is less than 3%, although there is a certain difference between the simulation results and the experimental results. Therefore, the simulation results are considered to reflect the experimental results realistically.

In order to investigate the correlation between the noise reduction effect and the geometric parameters *L*, *W* and *δ* at different angles, the OASPL test was statistically analyzed. Most statistical methods are based on normality assumptions. Therefore, it is necessary to perform a normality test on the OASPL before starting the analysis [25]. The normality test is carried out via Minitab. The result of the normality test (*p* > 0.05) indicates that the OASPL conforms to a normal distribution. In this research, a correlation analysis was carried out based on the multiple linear regression model of SPSS [26], and the results are shown in Figure 8. The abscissa represents the monitoring angle *θ*, and the ordinate represents the standard correlation coefficient beta. It can be seen that the OASPL is positively correlated with *L* and negatively correlated with *W* and *δ*. The groove depth has a greater impact on the noise reduction. However, the correlation between each parameter and the OASPL is inconsistent at different angles. The angles with the highest correlations for *L*, *W* and *δ* are 75°, 75° and 30°, respectively.

### 3.2. Analysis of Noise Generation

In order to analyze the causes of noise generation, a numerical simulation of the original nozzle was performed. The airflow inside the nozzle is complicated by the influence of the nozzle tube. However, variations in jet velocity and pressure are mainly concentrated at the nozzle exit. In order to simplify the model, we focused on the area at the exit of the nozzle tube. A relatively low velocity region is formed at the exit of the nozzle tube, due to its blocking. As shown in Figure 9, the flow field structure can be divided into four parts: the core region, inner mixed region, outer mixed region and the full-developed region [27,28]. Figure 10 shows the steady-state velocity and pressure of original nozzle in the symmetric cross section. The airflow is ejected from the slit, which is formed by the outer wall of the nozzle tube and the inner wall of the nozzle head. The pressure of the ejected air is higher than the ambient pressure, so an expansion sector develops at the exit of the nozzle tube, which leads to a gradual decrease in pressure. The expansion wave reduces the air pressure in the direction of flow and makes air expand and accelerate. When the expansion wave reaches the inner shear layer of the airflow, it is reflected to form a shock wave. The shock wave passes through the jet core region, reaching the outer shear layer of the airflow and reflecting to form an expansion wave [29,30]. As the airflow propagates downstream, the expansion and shock waves alternate, which leads to alternating low and high air pressure [31,32]. Due to the dissipative effect of the shock wave, the jet velocity decreases, and the shock wave structure gradually weakens and disappears. The two jets combine into a single jet and propagate downstream. Figure 11 shows the velocity profiles of different flow distances at the Y = 0 section. It can be seen that the velocity distribution of the two jets in the radial direction is not uniform, and on both sides of the jet there exists intense inner and outer velocity shear layers that correspond to the inner and outer mixed regions of the nozzle, respectively. As the airflow propagates downstream, the jet velocity gradually decreases and the blocking effect of the nozzle tube weakens, causing the inner shear layer to disappear and the two jets to combine.

The Q-criterion method has been widely used in vortex identification, which was proposed by Hunt in 1988 [33]. The iso-surface of the Q-criterion can visualize the vortex structure in the flow field. The Q-criteria iso-surface map of the nozzle section is shown in Figure 12. The 3D vortex structure is presented in Figure 12a. The airflow is ejected from the slit, resulting in a large velocity difference, which forms a shear layer. Due to the vigorous mixing of the original jet and the entrained flow, larger-scale vortex rings are formed in the inner and outer mixing regions. Large-scale vortices have been proven to generate stronger noise [34,35]. Due to the instability of the jet shear layer, vortex rings are torn and broken to form streamline vortices. As the motion proceeds, the vortices can break up and merge continuously, and the large-scale vortices gradually transform into small-scale vortices with weaker turbulence intensity, resulting in a reduction in noise emissions. There are many high-energy vortices at the exit of the nozzle tube (i.e., inside the shear layer) due to the large velocity changes, as shown in Figure 12b. The velocity and pressure change rapidly with the motion of the high-energy vortices, forming strong acoustical noise. As the jet propagates downstream, the surrounding air is continuously entrained, resulting in frequent diffusion and dissipation and energy reduction. Therefore, the flow fluctuations and gradient variation are greatly reduced, resulting in lower levels of acoustical noise. It can be determined from the above analysis that the acoustical noise is generated by the multi-vortex structures and the pressure fluctuations [36].

### 3.3. Effects of Micro-Groove Parameters

The parameters of micro-grooves have different effects on noise reduction, as shown in Section 3.1. Figure 13 compares the OASPL simulation results for different micro-groove parameters. The OASPL increases with the increase in *L* (as shown in Figure 13a), as well as decreases with increases in *W* and *δ* (as shown in Figure 13b,c). The velocity clouds for different groove parameters are shown in Figure 14. The length of the core region of the jet (i.e., the beginning of the groove to where the airflow begins to mix) is defined as the length of the high velocity region. The lateral distance of the velocity region 0.5 D from the nozzle exit is the width of the outer mixing region. Compared to the original jet (Figure 9a), micro-grooves can shorten the length of the high velocity region. The length of high velocity region increases with *L* and decreases with *W* and *δ.* It can also be noticed that the shortening of the high velocity region is accompanied with an increase in the width of outer mixed region. The effects of *L* and *δ* on the high velocity region are stronger than that of *W*. *δ* has the greatest effect. When *δ* is 0.5 mm, the high velocity region has the shortest length and the greatest width. This is consistent with the results of the regression analysis in Section 3.1. The reason why *L* and *δ* have a significant effect is that when *L* is smaller or *δ* is larger, more of the low velocity airflow at the exit of the nozzle tube is entrained (as shown in Figure 14a,i), resulting in enhanced jet mixing and accelerated velocity decay. It shows that enhancing the mixing of the jet at the nozzle exit can effectively suppress noise. This may be due to the fact that the enhanced mixing reduces the length of the high velocity region.

### 3.4. Analysis of Noise Cancellation Mechanism

#### 3.4.1. Analysis of Flow Field Characteristics

Micro-grooves have an excellent effect on reducing jet noise, as the results show in Section 3.1. Nozzle 7 was chosen for the analysis of the noise reduction mechanism. Figure 15 compares the centerline velocity of the micro-grooved nozzle and the original nozzle, where U_a_ is the velocity on the centerline and U_j_ is the mean velocity of the nozzle exit. The jet produced by the micro-grooved nozzle has a lower velocity and peaks at a shorter flow distance. The end of the outer mixing region is defined as the point where the nozzle has 90% of its maximum velocity [18]. The outer mixing region lengths are 6.36 D and 4.54 D for the original nozzle and the micro-groove nozzle, respectively, which indicates that the micro-grooves reduce the length of the jet mixing region.

Figure 16 shows the streamlines and velocity distributions of the original nozzle and the micro-grooved nozzle. It can be seen from Figure 16a,b that the micro-grooves significantly reduce the low velocity region at the exit of the nozzle tube. Moreover, the micro-grooves reduce the velocity fluctuations in the core area on both sides and shorten the core length significantly, despite expanding the outer mixing region. According to the streamlines in Figure 17a,b, a small vortex is formed at the bottom of the groove when the airflow passes through the groove. The disturbance generated by the vortices makes the airflow unstable in the axial and radial directions, increasing the mixing of airflow inside the nozzle. Similar conclusions are given by Jayant et al. [15,16]. More of the low velocity airflow at the exit of nozzle tube is entrained into the jet due to the vigorous mixing of airflow. The significant decrease in the low velocity airflow region results in a reduction of the internal recirculation vortices. The merging point of the jets is advanced to the nozzle exit, resulting in a shortening of the airflow mixing region. This is also illustrated by the turbulent intensity of the original nozzle and the micro-grooved nozzle, as shown in Figure 18. As the turbulent intensity develops downstream on the centerline, there exist two maximal values. The turbulence intensity of the micro-grooved nozzle at the nozzle exit increases significantly, but the distance between the two peaks is shortened from 7.6 D to 6 D. This indicates that the distance of the mixing region of the micro-grooved nozzle is shortened. The shorter mixing region reduces the spread of turbulence and effectively suppresses the noise.

#### 3.4.2. Vortex Analysis

Vortex sound theory reveals that large-scale vortex structures will generate noise during jet propagation [37]. Figure 19 compares the steady-state Q (Q = 1 × 10^11^) iso-surface at the same moment, which shows that the continuous large-scale vortex ring at the exit of the micro-grooved nozzle breaks into smaller vortices with lower energy (red circles in Figure 18). The evolution of the vorticity is shown in Figure 20. Formation of the axisymmetric vortex ring can be seen at the position X = 0 mm in the original nozzle. Then, the ring vortex begins to bend in the circumferential direction and lose stability, with vortical fragments beginning to show up at X = 4.2 mm. In the micro-grooved nozzle, due to the disturbances introduced by the grooves, the vortex ring loses its axial symmetry at X = 0 mm, and vortical fragments appear at X = 2.1 mm. With the jet continuing to propagate downstream, the vortex ring is further broken up at X = 4.2 mm. It also proves that the micro-grooves can enhance mixing of the jet, resulting in an acceleration of large-scale vortices breaking into small-scale vortices. The small-scale vortices have lower energy and generate less pressure fluctuations during the motion, resulting in lower noise. Meanwhile, although the disturbances in the micro-grooved nozzle are helpful for the fragmentation of large-scale vortices, the vortices at this position are stronger than those in the original nozzle. These results may explain the widening of the outer mixing region in Figure 16b.

#### 3.4.3. Analysis of Frequency Characteristics

From Section 3.1, it can be seen that *L*, *W* and *δ* have more significant noise reduction effects at 30° and 75°. Therefore, the 30° and 75° noise monitoring points were selected for frequency analysis, as shown in Figure 21. A micro-groove with different parameters changes the frequency spectra of aerodynamic noise. As can be seen in the figure, the micro-groove maintains the sound pressure level (SPL) of low-frequency noise. Furthermore, the variation of the *L*, *W* and *δ* reduce the SPL of both mid-frequency noise (500 Hz < *f* < 10 kHz) and high-frequency noise (*f* > 10 kHz). However, the effect in reducing the SPL of high-frequency noise is not satisfactory, and the reduction is only relatively obvious for 30°. Due to the directionality of the noise, the sound pressure level varies in different directions. The noise is mainly propagated downstream, so the high-frequency sound pressure level in the 75° direction is lower than that in the 30° direction. This results in a more pronounced reduction of the high-frequency sound pressure level in the 30° direction. Compared to *L* and *δ*, the change in *W* has a smaller effect on the SPL. It indicates that micro-grooves have the potential to reduce the SPL of mid-frequency noise and high-frequency noise. Therefore, the amplitude of the OASPL is reduced by controlling middle- and high-frequency noise.

## 4. Conclusions

In this study, the flow field and acoustical characteristics of an original nozzle and a micro-grooved nozzle were investigated using the hybrid RNAS/LES method and the FW–H method. The causes of noise generation from the original nozzle were analyzed, and the noise reduction mechanism of a micro-groove and the influence of its *L*, *W* and *δ* on noise reduction were discussed. The following conclusions can be drawn:By analyzing the results of velocity, pressure and the Q value in the simulation calculations, the reasons for jet noise were obtained: due to the blockage of the nozzle tube, the airflow forms a low velocity region at the exit of the nozzle tube. A large velocity difference forms between this region and the surrounding high velocity jet, resulting in the formation of vortices with a large structure and high energy in the inner and outer mixing regions of the jet. The movements of the vortices cause pressure fluctuations, and thus generate noise.Changes in the micro-groove parameters have an effect on the noise reduction. The OASPL decreases with increases in the *W* and *δ*, and increases with the increase in the *L*. The *δ* has the greatest effect on noise. The length of the jet’s high velocity region shows a similar variation rule. The nozzle with the best noise reduction was Nozzle 7, which reduced the OASPL from 97.36 to 90.7 dB at 30°, a reduction of about 6.66 dB.The noise reduction mechanism of a micro-groove was analyzed as follows. The jet flow through the micro-groove forms a vortex in the groove, which creates a disturbance to the original jet. These disturbances increase jet mixing, and cause destabilization of the inner shear layer of the jet, accelerating the development of the shear layer. The large-scale vortex ring is rapidly broken into smaller vortices with lower energy. The length of the high velocity region and the mixing region of the jet are reduced, although the turbulence intensity of the jet increases.The frequency analysis indicates that the micro-groove maintains the SPL of low-frequency noise and significantly reduces the SPL of mid-frequency noise. In addition, the SPL of high-frequency noise at specific angles is also reduced. Thus, the amplitude of the OASPL is reduced.

## Figures and Tables

**Figure 1 micromachines-14-01814-f001:**
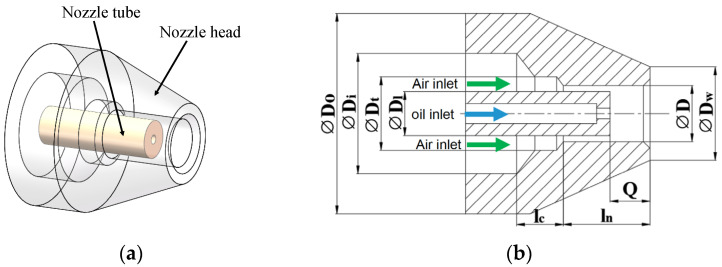
The structure of original nozzle. (**a**) Three-dimensional model. (**b**) Structural parameters.

**Figure 2 micromachines-14-01814-f002:**
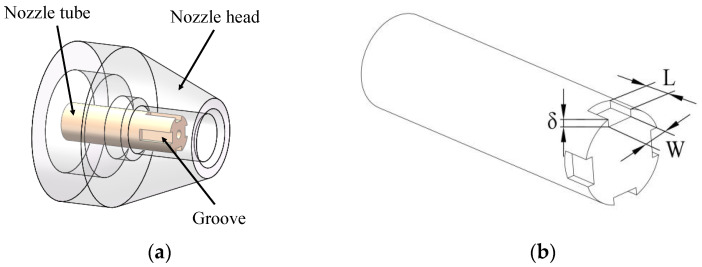
The structure of micro-grooved nozzle. (**a**) Three-dimensional model. (**b**) Groove structural parameters.

**Figure 3 micromachines-14-01814-f003:**
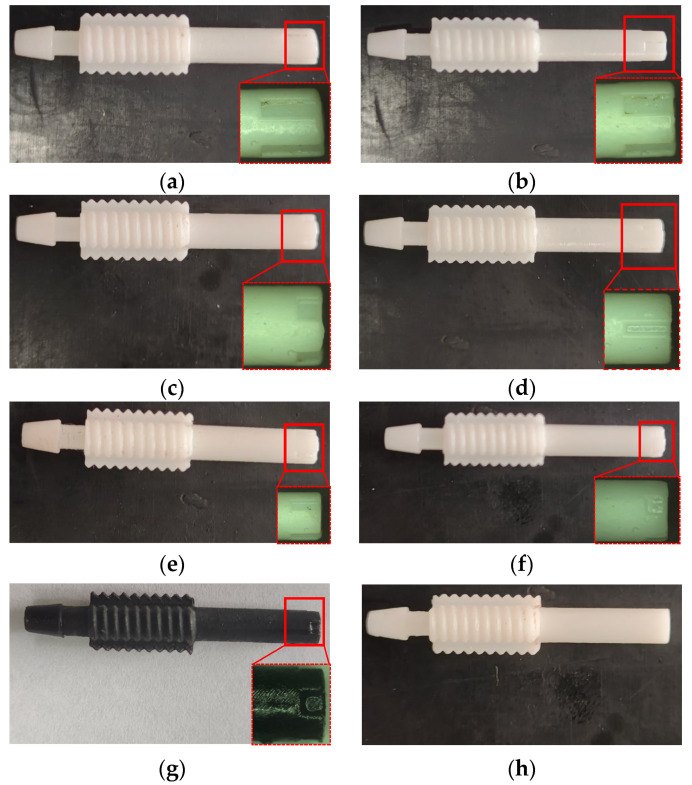
The actual nozzle tubes. (**a**) Nozzle 1 (*L*3 *W*1 *δ*0.35). (**b**) Nozzle 2 (*L*2 *W*1 *δ*0.35). (**c**) Nozzle 3 (*L*1 *W*1 *δ*0.35). (**d**) Nozzle 4 (*L*2 *W*0.5 *δ*0.35). (**e**) Nozzle 5 (*L*2 *W*0.7 *δ*0.35). (**f**) Nozzle 6 (*L*1 *W*1 *δ*0.2). (**g**). Nozzle 7 (*L*1 *W*1 *δ*0.5). (**h**) Original nozzle.

**Figure 4 micromachines-14-01814-f004:**
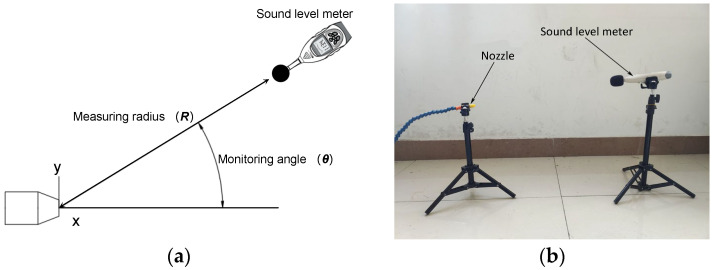
OASPL test. (**a**) Schematic diagram. (**b**) Experimental equipment.

**Figure 5 micromachines-14-01814-f005:**
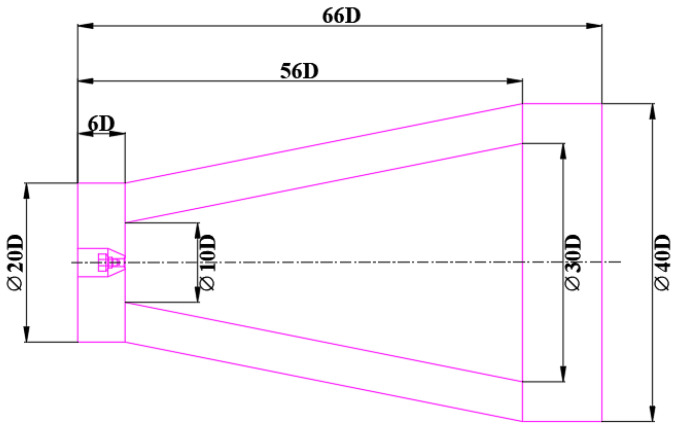
The size of the nozzle flow field.

**Figure 6 micromachines-14-01814-f006:**
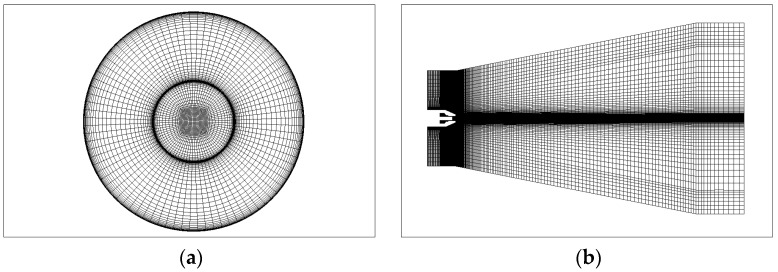
Flow field mesh. (**a**) Nozzle inlet section mesh. (**b**) Flow field axial section mesh.

**Figure 7 micromachines-14-01814-f007:**
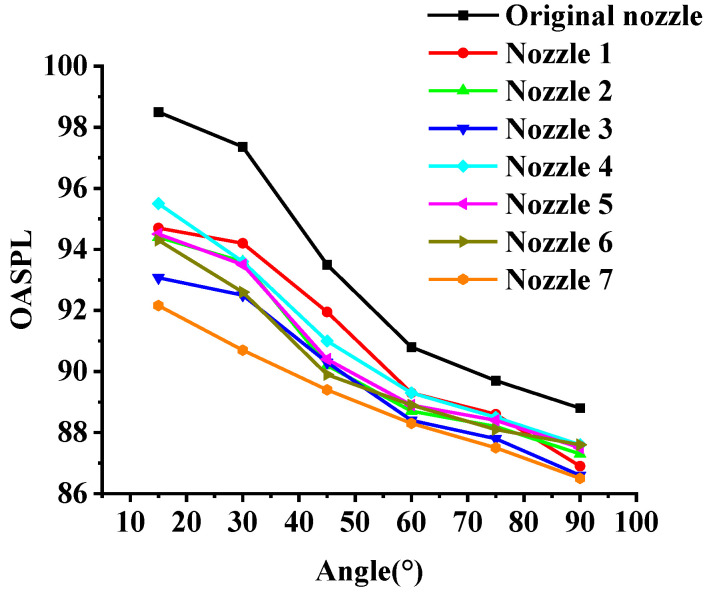
The experiment results of OASPL.

**Figure 8 micromachines-14-01814-f008:**
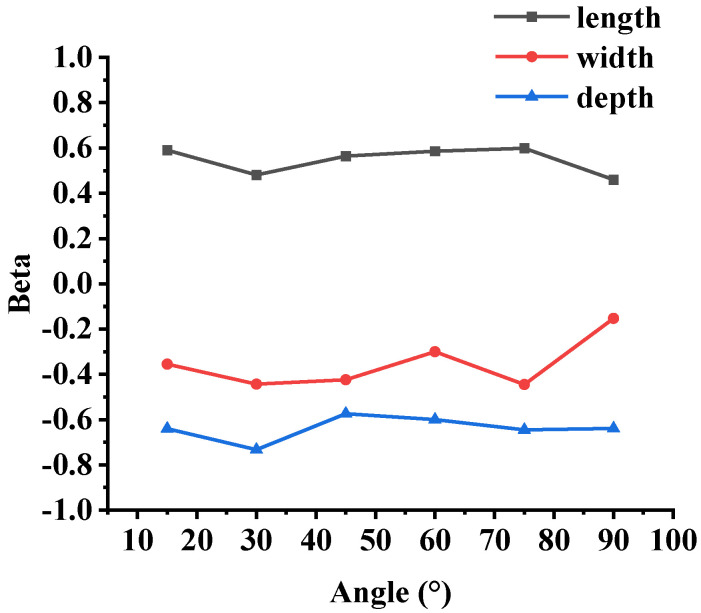
Standard beta correlation coefficients of experiment results.

**Figure 9 micromachines-14-01814-f009:**
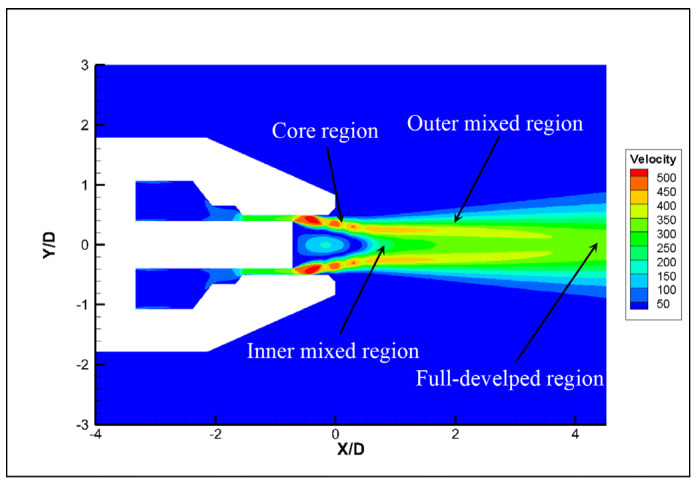
Jet flow characteristics.

**Figure 10 micromachines-14-01814-f010:**
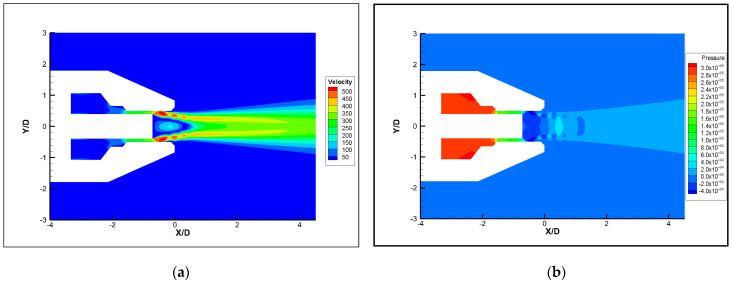
Velocity and pressure contours at steady state (Z = 0). (**a**) Local velocity contour. (**b**) Local stress contour.

**Figure 11 micromachines-14-01814-f011:**
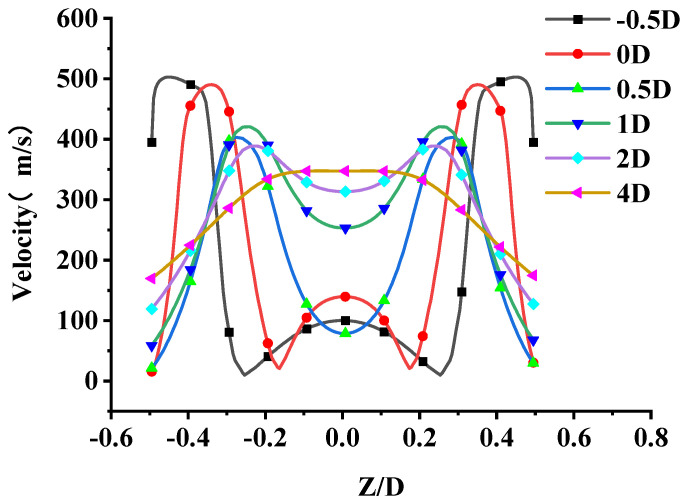
Radial velocity of nozzle at different distances (the axisymmetric plane).

**Figure 12 micromachines-14-01814-f012:**
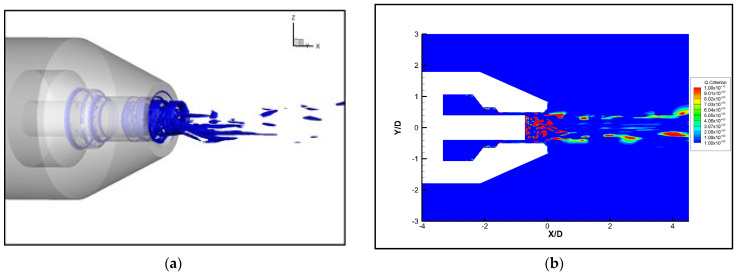
Local Q-value cloud map. (**a**) 3D vortex structure. (**b**) Q-value cloud map.

**Figure 13 micromachines-14-01814-f013:**
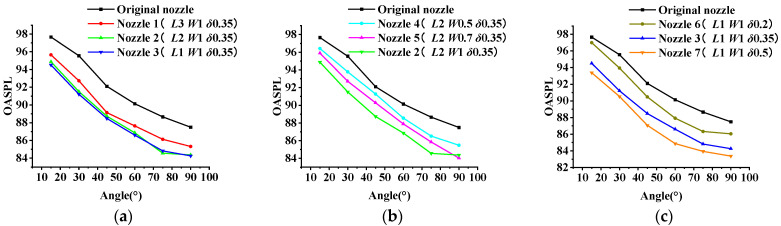
The OASPL simulation results for different micro-groove parameters. (**a**) The influence of *L.* (**b**) The influence of *W.* (**c**) The influence of *δ*.

**Figure 14 micromachines-14-01814-f014:**
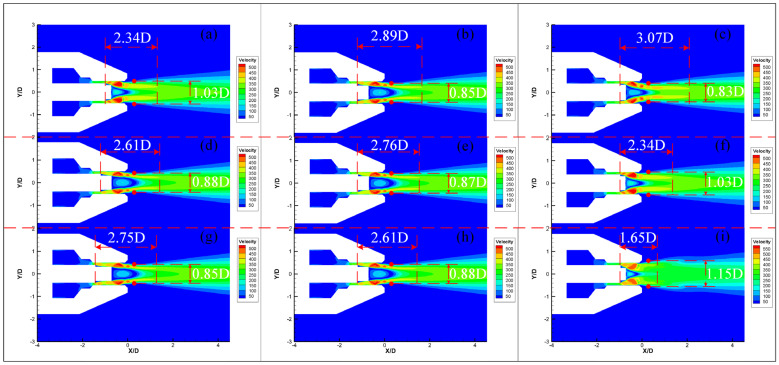
The velocity clouds for different micro-groove parameters. (**a**) Nozzle 3 (*L*1 *W*1 *δ*0.35). (**b**) Nozzle 4 (*L*2 *W*0.5 *δ*0.35). (**c**) Nozzle 6 (*L*1 *W*1 *δ*0.2). (**d**) Nozzle 2 (*L*2 *W*1 *δ*0.35). (**e**) Nozzle 5 (*L*2 *W*0.7 *δ*0.35). (**f**) Nozzle 3 (*L*1 *W*1 *δ*0.35). (**g**) Nozzle 1 (*L*3 *W*1 *δ*0.35). (**h**) Nozzle 2 (*L*2 *W*1 *δ*0.35). (**i**) Nozzle 7 (*L*1 *W*1 *δ*0.5).

**Figure 15 micromachines-14-01814-f015:**
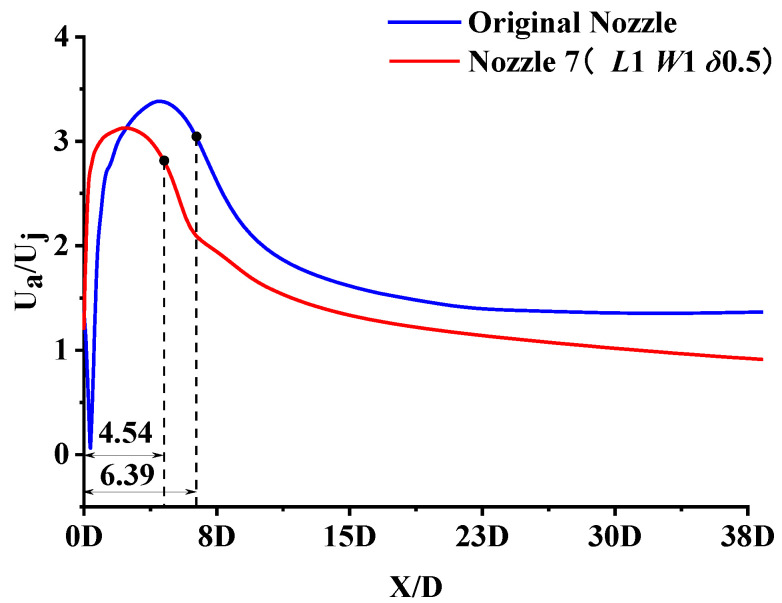
Comparison of dimensionless centerline velocity distributions.

**Figure 16 micromachines-14-01814-f016:**
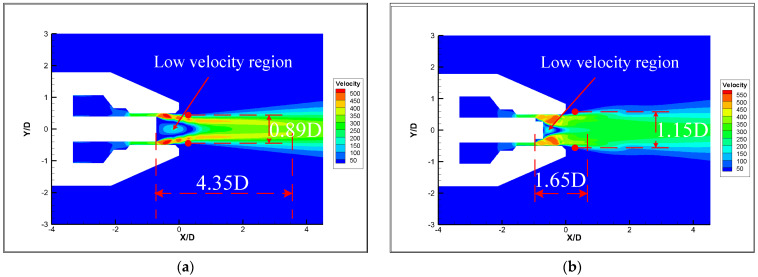
Velocity distributions. (**a**) Original nozzle. (**b**) Nozzle 7.

**Figure 17 micromachines-14-01814-f017:**
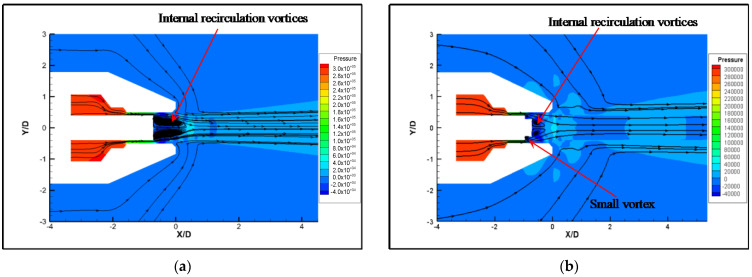
Streamline distributions. (**a**) Original nozzle. (**b**) Nozzle 7.

**Figure 18 micromachines-14-01814-f018:**
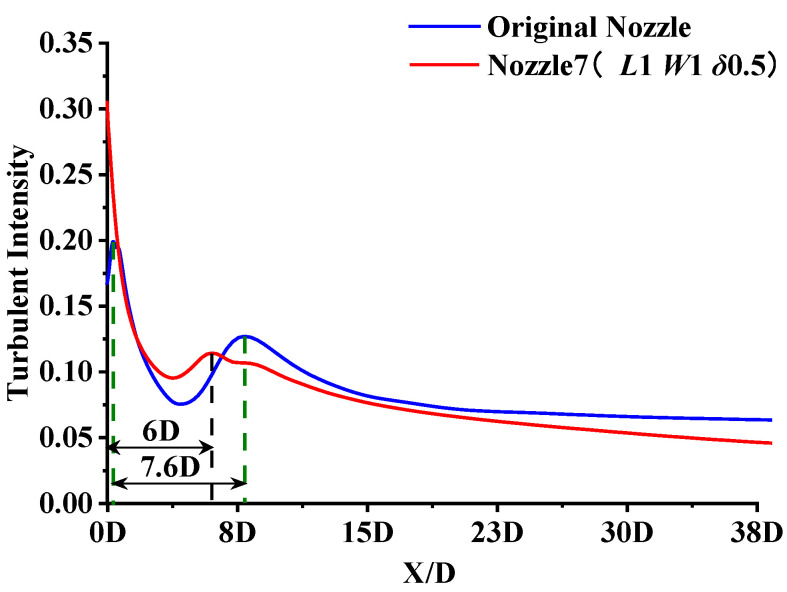
Comparison of centerline distributions of turbulent intensity.

**Figure 19 micromachines-14-01814-f019:**
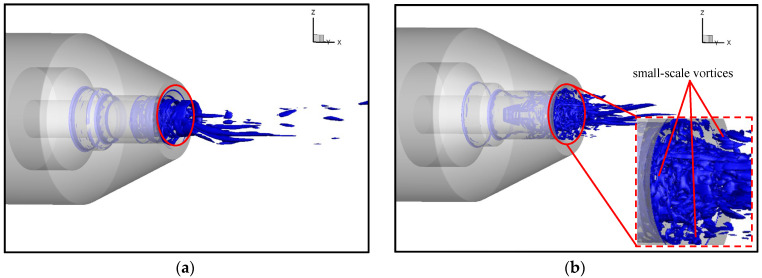
Iso-surface of Q value. (**a**) Original nozzle. (**b**) Nozzle 7.

**Figure 20 micromachines-14-01814-f020:**
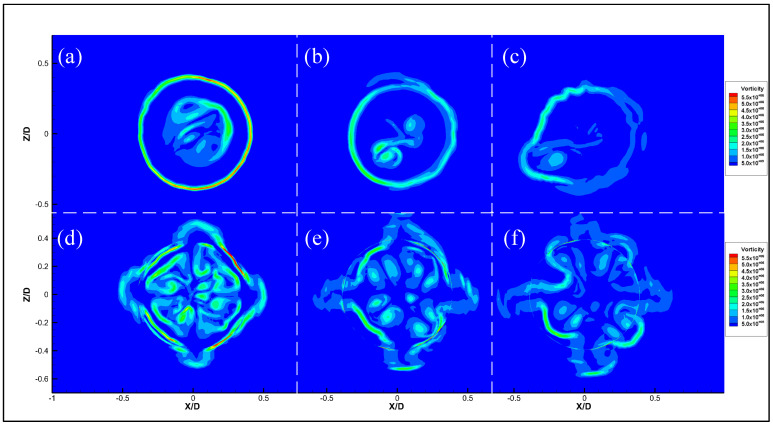
Development of the ring vortex along the jet axis. (**a**) Original Nozzle, X = 0. (**b**) Original Nozzle, X = 2.1. (**c**) Original Nozzle, X = 4.2. (**d**) Nozzle 7, X = 0. (**e**) Nozzle 7, X = 2.1. (**f**) Nozzle 7, X = 4.2.

**Figure 21 micromachines-14-01814-f021:**
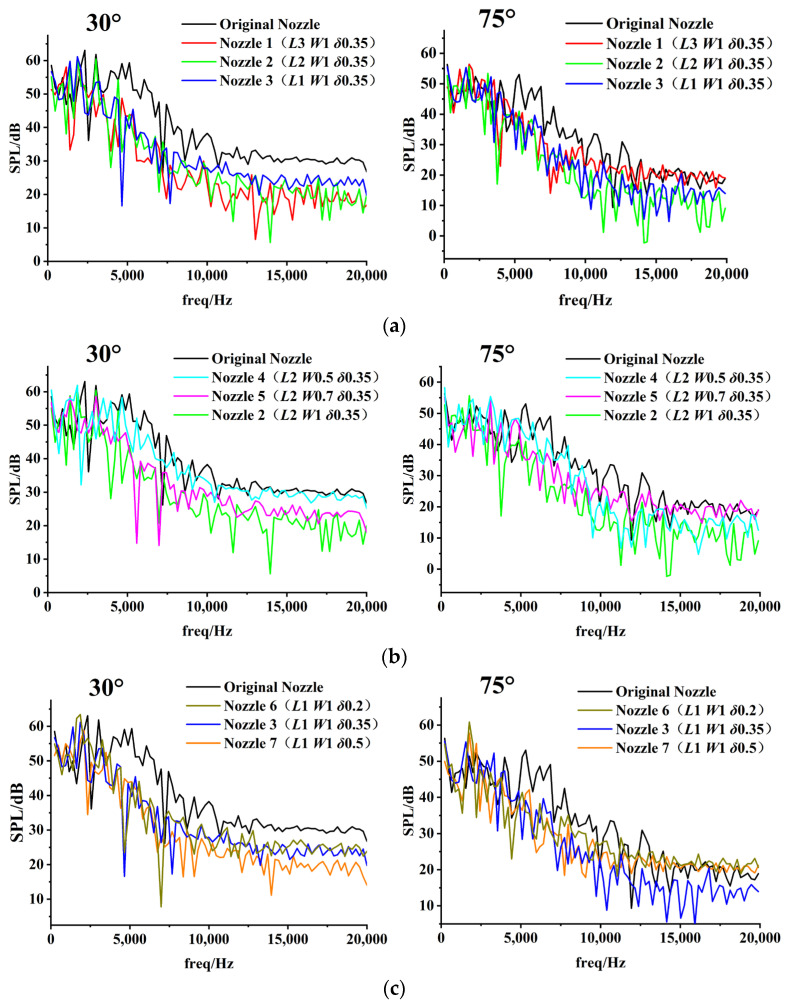
Influences of micro-groove parameters on frequency spectra. (**a**) Influence of *L*. (**b**) Influence of *W*. (**c**) Influence of *δ*.

**Table 1 micromachines-14-01814-t001:** Parameters of the original nozzle.

*D_o_*/(mm)	*D_i_*/(mm)	*D_l_*/(mm)	*D_t_*/(mm)	*D*/(mm)	*D_w_*/(mm)	*l_c_*/(mm)	*l_n_*/(mm)	Q/(mm)
15	9	3.3	5.5	4.2	7	3.5	6.5	3

**Table 2 micromachines-14-01814-t002:** Parameters of different combined nozzles.

Number	*L*/(mm)	*W*/(mm)	*δ*/(mm)
Nozzle 1	3	1	0.35
Nozzle 2	2	1	0.35
Nozzle 3	1	1	0.35
Nozzle 4	2	0.5	0.35
Nozzle 5	2	0.7	0.35
Nozzle 6	1	1	0.2
Nozzle 7	1	1	0.5

**Table 3 micromachines-14-01814-t003:** The OASPL analysis with different grid numbers.

Grid Numbers	Maximum OASPL Deviation (dB)	Average OASPL Deviation (dB)	Mean Relative Deviation (%)
227W	4.22	2.536	2.734
289W	1.31	0.827	0.890
367W	5.113	3.071	3.311
459W	3.155	1.627	1.754

**Table 4 micromachines-14-01814-t004:** Error analysis of OASPL simulation results for micro-grooved nozzle.

Nozzle Type	Maximum OASPL Deviation (dB)	Average OASPL Deviation (dB)	Mean Relative Deviation (%)
Original nozzle	1.31	0.827	0.89
Nozzle 1	2.81	1.84	2.02
Nozzle 2	3.67	2.101	2.32
Nozzle 3	2.98	1.952	2.17
Nozzle 4	2.12	1.0225	1.12
Nozzle 5	3.54	1.59	1.76
Nozzle 6	2.62	1.483	1.64
Nozzle 7	3.56	2.308	2.59

## Data Availability

Data will be made available on request.

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
