# Peer review of "Minimum Quantity Lubrication Jet Noise: Passive Control"

_micromachines, 2023, doi:10.3390/mi14101814_

Round 1

Reviewer 1 Report

In this work, the authors proposed a micro-groove nozzle to reduce jet noise of MQL spray. The noise generation mechanism of this jet noise and the performance of the micro-groove nozzle were numerically investigated. Authors demonstrated numerically the proposed design of nozzle can speed up the process of breaking down large vortex and enhance the mixing of airflow. This work also studied the effect of geometric parameters of the micro-groove on the noise reduction performance. Although this work presents some interesting results, the following issues need to be addressed before acceptance can be considered:

1.     This kind of groove structure have been used in the literature with proven performance in reducing flow-induced noise. The innovative of the work should be further explained.

2.     Authors have studied the effect of geometric parameters (L, W and σ) on the noise reduction performance. In Section 2.2, a brief explanation of why those values of the parameters were chosen in this study has been discussed. However, it is a bit too general to say that when L is too large (or when L is too small), then what would happen. Would there be a range of those values relative to the original geometry of the nozzle? In other words, L=1mm, 2mm, 3mm were chosen. Does it depends on the geometry of the nozzle, like ln or lc+ln? If a nozzle with longer ln (or lc+ln) is considered, is the value of L be the same? Similar for W and σ.

3.     As indicated in Section 2.3, a number of OASPL experiments has been conducted. However, such experimental results were not shown in the article. Including this experiment results can let the readers have a brief idea of the directivity of the acoustic pressure radiated by the nozzle.

4.     Table 3 need revision. Chinese wordings appear in the first row of the table.

5.     Regarding the results shown in table 3, why the deviations are not gradually reduced wit the increase of grid numbers? It is not convincing that the numerical results are convergent by looking at the results shown in table 3.

6.     How much does the airflow change compared to the original nozzle?

7.     It is a bit difficult to see the lines and words in Figure 12.

There is just minor editing of English language is required. Overall, it is easy to understand and follow. However, there are some Chinese wordings appeared in the table 3. 

Author Response

Thank you very much for your suggestions. We have responded point by point.

Reviewer 2 Report

The manuscript proposed a method to reduce jet noise for MQL nozzles. The noise reduction is based on micro-groove noise reduction concept. The flow field and acoustical characteristics have been studied. This work reveals the effect of groove shape parameters on the jet noise. Simulation has been done to verify the experimental results. Corresponding mechanism of noise reduction has been proposed. Major revision is suggested. There are a few suggestions that the author may consider:

1. Regarding the nozzle design, how was the shape parameters range selected? The discussion in 2.2 is very general. More details and references are needed.

2. The inlet air pressure through a jet nozzle may vary depending on specific application. Would the micro-groove parameter effect be more, less or equally significant with a higher air pressure?

The manuscript needs to be revised for language and spelling issue. (e.g. sentence is missing in line 89)

Author Response

(The authors gave the same response as above.)

Reviewer 3 Report

1.The second sentence in section 2.2 is not to be understood. “ And Fig. 3 provided Since micro- grooves facilitate noise control, four grooves were designed at the end of nozzle tube with uniform distribution.”

2.The authors investigated the effect of the groove parameters ( L, W and δ) on the noise reduction performance. In Section 2.2, the basis of the parameter values is described in a general way. Whether there is also some relationship between these values and the original nozzle geometric parameters?

3.The title of table 3 needs to be revised. It is a Chinese expression.

4.According to the results in Table 3, why the deviation in noise does not decrease with the number of grids? Does this indicate that the numerical simulations did not converge?

5.Generally, acoustic experiments need to be carried out in an anechoic chamber to eliminate external interference. Why didn't the authors perform the noise experiments in an anechoic chamber?

6.In Figure 14, the authors numerically compare the length of the jet core between different groove parameters. And, how much the jet core length has changed compared to the original nozzle?

7.The lines in some pictures are too thin, which affects the readability. It is recommended to bold them.

8.The display of the text in Figure 13 is not obvious, which causes some trouble for reading.

no

Author Response

(The authors gave the same response as above.)

Round 2

Reviewer 1 Report

Authors have addressed issues suggested from the first review and the quality of paper has been improved.

Reviewer 2 Report

The authors clearly addressed my comments. The manuscript was much improved.